# Admission to the Long-Term Care Facilities and Institutionalization Rate in Community-Dwelling Frail Adults: An Observational Longitudinal Cohort Study

**DOI:** 10.3390/healthcare10020317

**Published:** 2022-02-07

**Authors:** Susanna Gentili, Fabio Riccardi, Leonardo Emberti Gialloreti, Paola Scarcella, Alessandro Stievano, Maria Grazia Proietti, Gennaro Rocco, Giuseppe Liotta

**Affiliations:** 1Department of Biomedicine and Prevention, University of Rome “Tor Vergata”, 00133 Rome, Italy; fariccardi@hotmail.com (F.R.); leonardo.emberti.gialloreti@uniroma2.it (L.E.G.); paola.scarcella@uniroma2.it (P.S.); giuseppe.liotta@uniroma2.it (G.L.); 2Centre of Excellence for Nursing Scholarship, OPI, 00192 Rome, Italy; alessandro.stievano@gmail.com (A.S.); mg_proietti@alice.it (M.G.P.); genna.rocco@gmail.com (G.R.)

**Keywords:** assisted living facilities, residential facilities, frail older adult, nursing homes, long-term care, admission rate, multidimensional frailty, community-dwelling older adults, institutionalization, nursing home placement

## Abstract

The worldwide aging and the increase of chronic disease impacted the Health System by generating an increased risk of admission to Long-Term Care (LTC) facilities for older adults. The study aimed to evaluate the admission rate to LTC facilities for community-dwelling older adults and investigate factors associated with these admissions. A secondary data analysis stemming from an observational longitudinal cohort study (from 2014 to 2017) was performed. The sample was made up by 1246 older adults (664 females and 582 males, mean age 76.3, SD ± 7.1). The LTC facilities access rate was 12.5 per 1000 observations/ year. Multivariable Linear Regression identified frailty, cardiovascular disease, and incapacity to take medicine and manage money as predictors of the LTC facilities’ access rate. The Multiple Correspondence Analysis identified three clusters: those living at home with comorbidities; those living in LTC facilities who are pre-frail or frail; those very frail but not linked to residential LTC. The results indicate that access to LTC facilities is not determined by severe disability, severe comorbidity, and higher frailty levels. Instead, it is related to moderate disability associated with a lack of social support. Therefore, the care policies need to enhance social interventions to integrate medical, nursing, and rehabilitative care.

## 1. Introduction

The worldwide rise of older adults, the life expectancy, and the increase of chronic diseases requires the (re)configuration of the health systems [1,2]. Indeed, the ensemble of these phenomena impacted the system by also generating an increased risk of admission to Long Term Care (LTC) facilities for older adults. Although it was widely discussed that older people prefer to stay in their families’ home environment to reduce autonomy loss and preserve the quality of life [3,4], recently, there have been some significant changes in the use of LTC facilities. 

In this context, some studies have investigated the risk factors of admission LTC facilities, analyzing the socio-demographic characteristics, the clinical condition, and the dependence level on performing the activity [5,6,7]. For example, it was widely known internationally that older age [8,9], gender [10], and social support [7,11,12] were significant in predicting the risk of institutionalization. Furthermore, among various risk factors investigated, were reported the role of comorbidity and functional disability [5]. However, in 2010, the systematic review of Luppa et al. has analyzed the clinical conditions associated with LTC facilities placement. The authors have found insufficient evidence about the comorbidities’ role as a predictor of institutionalization [5]. Despite this review, several studies have focused on the association between central nervous system diseases (i.e., Parkinson’s, Dementia, and Stroke) and the risk of LTC facilities admission [5,7,13,14,15]. 

In recent years an increasing number of relevant studies focused on the frailty and significant frailty adverse health outcomes (i.e., mortality [16,17], hospitalization [18], and institutionalization [19]). The frailty in the bio-psycho-social model has been described as a dynamic state that can induce the loss of one or more domains (clinical physical, cognitive, psychological, functional, social, and economic) [20]. Given its dynamic nature, frailty may rapidly progress to disability and thus increase the risk of adverse events. However, to our knowledge, only two reviews have analyzed the relationship between frailty and institutionalization [21,22]. The first one, by Wang et al., analyzed the predictive role for nursing home admissions, finding that only two included articles had shown the significant predictive role of frailty [22,23,24]. On the other hand, Kojima’s systematic review and meta-analysis included five studies that confirmed frailty as a significant predictor of nursing home admission among community-dwelling older people [21,23,25,26,27,28]. 

In addition, it was found that there is a severe lack of knowledge in the scientific literature regarding access rates to LCT facilities. Therefore, determining the access rate and determinants of LTC facilities’ admission is a matter of interest to policymakers, researchers, older adults and their families, and related parties, including General Practitioners (GPs), community nurses, social workers, and clinicians. Thus, the current study aims to evaluate the admission rate to LTC facilities for community-dwelling older adults, to give accurate data of a representative sample of Lazio Region (Italy), and investigate the factors associated with the admission to LTC facilities.

## 2. Materials and Methods

### 2.1. Study Design 

A secondary analysis of an observational longitudinal cohort study that analyzed the frailty level of older adults was performed. The primary purpose of this second study was to evaluate the access to Long Term Care (LTC) Facilities of frail older adults.

The study, which complied with the ethical standards of the 1965 Declaration of Helsinki, was approved by the Independent Ethical Committee of the University of Rome “Tor Vergata” (registration number: 95/15). Moreover, all participants in the study signed the informed consent form. 

The survey [29], and follow-up [18], have been described and published elsewhere. Information relevant to this study is described below

### 2.2. Sample Collection and Sample Size

In order to gather a representative sample of the population resident in the Lazio Region (Italy), sample collection at the baseline was carried out, in two phases:1.First phase: A group of general practitioners (GPs) was selected by block randomization from the Local Health Authorities (LHA) archives.2.Second phase: another randomization to select 25 patients over 64 years was performed by each GPs.

The Italian LTC facilities included the Nursing Homes (NH where social and health care are provided to partially or not-self-sufficient patients) and the Assisted Living facilities (AL where only social care is provided to individuals who should be self-sufficient/partially self-sufficient at least at the time of the admission). The follow-up about the admission to the LTC facilities was carried out after three years. Three years later data collection was conducted from the regional health database of Lazio to detect accesses to the NH and by querying GPs involved in the study to learn about accesses to AL facilities, which are not recorded by any database. The sample consists of 1246 individuals aged more than 64 years. During the 3-year follow-up, 22 people died before recording the follow-up collection, and it is unknown whether they have been admitted to LTC facilities.

### 2.3. Measurement 

The questionnaires used for the baseline collection were:•The Functional Geriatric Evaluation (FGE) questionnaire [30] assesses the multidimensional bio-psycho-social frailty in older adults. The FGE was the Italian version [29,30] of Grauer functional rating scale [31] validated in 2005 by Palombi et al. [32]. This questionnaire analyzes five domains, physical health, mental health, functional state, social resources, and economic resources, contributing to the final synthetic score (FSS). The FSS identified four levels of frailty: Very Frail, Frail, Pre-Frail, and Robust, with a score of ≤10, a score of >10 but <50, a score of ≥50 but ≤70, and a score of >70, respectively. The final score ranges from 108 to −101.•A list of 21 diseases assessed, with the support of GPs, to evaluate the presence or absence of comorbidity. The presence of comorbidity was considered when there were two or more active disease.•Activities of Daily Life (ADL) [33] and Instrumental Activities of Daily Life (IADL) [34] questionnaires that were administrated to define the level of disability to each participant in the study. Moderate disability was defined by dependence in performing IADL while severe disability was defined by dependence in performing ADL.•The absolute number of accesses to nursing facilities or home care has been retrieved from the Regional Health Database and the GPs for each participant involved in the study.

### 2.4. Statistical Analysis

The statistical analyses were carried out with IBM SPSS Statistics version 25.0 (IBM^®^ SPSS^®^ Statistics, Chicago, IL, USA). The primary outcome of this analysis was to calculate, for each participant, the LTC facilities access rate per 1000 observations/year.

The socio-demographic data, such as age (<74 years, between 75 and 85 years, and >85 years), gender, education (no education, primary school, secondary school, university/degree), living arrangements (alone, couple, other relatives, homeworker) comorbidities (more than two diseases vs. 0–1), bio-psycho-social frailty, disability (none, moderate, severe) of the sample were analyzed using descriptive statistics, such as mean, frequencies, percentage, and standard deviation.

Moreover, the access rate was stratified for frailty level (Robust, Pre-Frail, Frail and Very Frail) and Social Support to understand the relationship. 

Lack of Social Support has been defined as the presence of three conditions: living alone, lack of supporting neighbors, and lack of the possibility of receiving occasional help as delivering food or drugs at home in case of need. The last two conditions were self-declared by respondents (recorded by the FGE questionnaire). In addition, the number of respondents who accessed the Emergency Room (ER) over the 3-years follow-up as well as the ones who have been admitted to the hospital have been analyzed to explore the relationships with the LTC accesses 

To deepen the analysis of factors associated with institutionalization, every single variable in the questionnaire, as well as each ADL according to Katz and IADL according to Lawton, have been matched with the LTC facilities access (dichotomous variable, Yes vs. No) and the Chi-Square was performed to evaluate the association. The variables that showed a statistically significant association with the LTC access (*p* < 0.05) had been included in a multivariable linear regression model.

Finally, a Multiple Correspondence Analysis (MCA) [35] was performed in order to explore the mutual relation among disability, frailty, comorbidity, education, gender and age. There were utilized some criteria to define the number of dimensions to use: (i) scree test [36]; (ii) the eigenvalue [37]; (iii) Cronbach’s alpha score [37]. Moreover, as these criteria suggested, we chose the solution with two dimensions. Cronbach’s alpha score of less than 0.80 was considered acceptable because this is an exploratory analysis, and there was a poor correlation between the constructs [35,38].

## 3. Results

The socio-demographic characteristics of the final sample (N = 1246) are shown in Table 1.

The sample was predominantly female (53.2%), and the average age at baseline was 76.25 ± 7.129. Thus, patients mainly belonged to two age groups: less than 74 (44.1%) and between 75 and 85 years old (41.7%). Most of the patients lived with their spouse or child, respectively, 52.1% and 21.8%. A large part of the sample lived alone (20.7%). The education level shows that 47.1% of the sample completed primary school, 24.9% middle school, 15.3 % high school, and only 4.9% achieved a degree. Based on the questionnaire score, the sample is divided into four classes, respectively, Robust (43.7%), Pre-Frail (36.3%), Frail (13.2%), and Very Frail (6.8%). The prevalence of comorbidity was 81.7% (more than one disease). Moreover, 23.7% were moderately disabled (unable to perform at least one IADL), and 6.2% were severely disabled (unable to perform at least one ADL). 

The socio-demographic characteristics of interviewed lost at Follow-Up (see Table A1) differ from the total sample. Twenty-two people were lost at Follow-up because they died, and the GPs did not know whether they accessed LTC facilities. There were differences in age group distribution, mean and standard deviation, and many lived with a paid assistant. Moreover, the sample has a higher level of frailty.

The rate of access to the nursing facilities was 12.5 per 1000 person/year. The facilities access rate per person/year stratified for the age group was 8.4, 13.1, and 23.3 per 1000 observations/year, for 65–74, 75–84, and >85 years, respectively.

The analysis of access rate stratified for frailty level (Table 2) shows that frail older adults had the highest LTC facilities admission rate (28.4 per 1000 observation/year) followed by the very frail (18.1 per 1000 observation year). Patients with comorbidities also showed a higher LTC access rate than the others (14.5/1000 p/y vs. 3.6/1000 p/y). 

Figure 1 shows the distribution of the Frailty FSS for community-dwelling older people, residents in a NH and residents in AL facilities.

The boxplot (Figure 1) highlighted a lack of significant differences of the FSS score across the three settings, especially for men. The number of interviewed with very low FSS living at home, supported mainly by the families, is worthy of note.

To describe the relation between bio-psycho-social frailty and residential LTC, absolute numbers and percentage of LTC facility accesses stratified by level of frailty and presence or absence of social support are reported (Table 3). The table showed a bi-phasic relationship of Social Support and institutionalization. 

In fact, close to 50% of respondents who accessed the LTC facilities during the follow up were Robust (17 out of 39) with satisfactory social support (16 out of 17, 94.1%). On the opposite, among the other groups with a higher level of frailty, the lack of social support was associated with access to LTC facilities. 

At the same time, only one third of people who accessed LTC facilities showed a functional impairment at the baseline (13/39, 33.4%). The hospitalization rate in the period before entering the LTC was higher for persons who entered LTC facilities (6.3% vs. 2.8%, *p* = 0.010) as well as the Emergency Room access rate (5.4% vs. 2.4%, *p* = 0.026). However, 26/39 individuals entered the NH without hospital admission, and 18/39 without access to the Emergency Room.

It is noteworthy to point out that in absolute numbers, non-frail individuals (robust + pre-frail ones) who accessed LTC facilities were the majority (27/39). 

A multivariate linear regression model was carried out to test if the variables selected by the univariate analysis based on the CHI-Square test, significantly predict LTC facilities’ access rates (Table 4). The results showed the relevance of cardiovascular diseases as predictors of higher LTC facilities’ rates. In addition, among ADL and IADL, we found that functional incapacity to take drugs and manage payments were associated with higher access rates. Moreover, the inhabitants (i.e., the number of residents in the city where the data were collected) were a significant predictor of higher nursing facilities’ rates. Comorbidity, disability, hospital admission rate and ER access rate have not included in the final model because they were not statistic significant in the multivariable analysis, even if they have been selected by the univariate analysis.

Finally, the higher was the FSS, the higher was the access rate, underlining a surprising outcome: people robust, pre-frail, and Frail were at higher risk of entering a residential facility or a nursing home than the Very Frail one.

The joint category plot of the explored variable categories is shown in Figure 2. The facility’s use explained a portion of inertia (0.013 for the first dimension and 0.017 for the second dimension). The most discriminant variables for dimension 1 were FSS (0.813), disability (0.794), age group (0.425), and comorbidity (0.274), while the most discriminant variables for dimension 2 were FSS (0.732) and disability (0.694). In dimension 1, comorbidity is significantly correlated with disability (0.315) and FSS (0.268) and had a poor correlation with Facilities use (0.11); in dimension 2, the main correlation with the disability was FSS (0.765), and age group (0.315). From the graphic representation emerged that there were three main clusters. The first one, including most of robust and pre-frail individuals, involved those who lived at home, with no disability but with comorbidity and growing age. The second one included those who lived in-LTC facilities, had a moderate disability, and were classified mainly as Frail. Finally, the third one involved those with a higher level of frailty and a severe disability without a strong correlation with the facilities use.

## 4. Discussion

The nature of frailty as predictor of adverse health outcomes (i.e., hospitalization, mortality, and institutionalization) in a population who is aging, and with increasing prevalence of chronic disease is widely discussed [17,18,21,39,40]. Therefore, this study is aimed to assess the predictors of LTC facilities accesses, during three years following a frailty assessment. Our results mainly agree with those reported in previous studies [5,21,41].

First, our results have shown that the LTC facilities access rate was 12.5 per 1000 person/year. For the first time to our knowledge, we know the cumulative LTC access rate that is the sum of NH plus AL facilities access rate; this allows to plan interventions based on the real demand for community care that results from information not included in routine data flows but need to be gathered with ad hoc studies [17,23,25,26,27,28]. Moreover, from our results, if we analyze the rate stratified by frailty level, there is an evident expression of the health care market, where offer creates demand. This phenomenon is represented by the Robust and Pre-Frail that had access to the LTC facilities even if they did not show any functional impairment at least at the baseline, only because there are more places than the Very Frail and Frail individuals need. According to previous studies, Frail and Very-Frail older adults tended to stay at home even with medium to severe health conditions and low social support or limited social network [39,40,42,43]. According to our results, the LTC facility access rate is highest for Frail and Very Frail individuals. However, the Robust and Pre-Frail individuals who access LTC facilities are 70% of the total accesses due to the limited percentage of Frail and Very Frail individuals in the over65 population. However, as shown in previous systematic reviews [21] pre-frailty is significantly associated with increased LTC facilities access [23,25,26]. These results point out the need to (re)configure the LTC system increasing the offer of home care [18,29] thus reducing the number of LTC accesses generated by individuals with limited or without functional impairment. 

The multivariable analysis results showed that factors such as disability, social support, hospital admission rate, ER access rate, and comorbidity were not associated with an increase in the LTC access rate. This is probably because half of the ones who accessed residential LTC were robust individuals enjoying satisfactory social support. In these cases, we cannot identify the mechanisms that lead to institutionalizations. 

Analyzing sample-specific factors such as gender, age and living arrangements, we have found they are not significantly associated with LTC access. This data is in contrast to some recent studies. For example, in 2015, Hajek et al. had demonstrated that being widowed was a major longitudinal risk factor of institutionalization [11]. The systematic review of Luppa et al. [5] identified the predisposing variables with solid evidence: increased age [8,44] and not owning a home [5,45,46]. Moreover, they found predisposing variables with moderate evidence: unmarried or widowed and older adults with low social support [5,39,40,43]. On the contrary, we did not find an association between the admission and the marital status, maybe due to a limited number of those who had access to LTC facilities. 

More studies are focused on dementia [11,14,15,39] and found a significant and positive association with LTC access. However, in contrast with this, we have found that one of the predictors of LTC access was cardiovascular disease, the most prevalent disease. Other studies [5,21,47], found that disability as a whole, and functional impairment influence the LTC access as predictors. It is not the case of this study that found only two activities such as being able to take therapy properly and managing money independently were associated to higher LTC facilities access rate. The functional impairment in performing ADL and IADL is at the same time consequence of increased frailty and age and cause of worsening of frailty status, thus not independent from frailty itself in predicting LTC access [21]. Indeed, seeing the tendency to stay at home, the main reason for accessing residential LTC among pre-frail and frail individuals seemed to be an imbalance between functional impairment in the performance of ADLs and IADLs and the lack of caregivers supported by home health and social care services. This is also evident from the MCA results, which showed the association between LTC facilities and moderate disability with a moderate level of frailty. Our results showed that, despite high levels of frailty and poor social support, the elderly population tends to stay at home as much as possible, looking for a balance between needs of care and available support, even if associated to a poor quality of life. 

That is due to the complexity of this phenomenon, which may be influenced by social, economic, and cultural features related to the country where it is analyzed [21]. As a consequence of this complexity due to the multi-factorial influence, we have found that living in a big city increases the risk of institutionalization compared to a small town, perhaps thanks to the more availability of LTC structure or the poor livability in the city [45,46]. 

One of the main limitations of this study is that maybe some older adults during the three years have access to the LTC facilities but die almost immediately. For this reason, the limitation is linked to the people we may have lost due to missed reports from GPs. A second limitation is the lack of information about the trajectory of frailty, and especially about component of frailty due to the functional status, during the three years of follow up. However, most of the ones who entered the LTC facilities did not need to be admitted to the hospital before, so it is likely that physical decline did not play a major role in their decision of moving from their homes to the facilities. Moreover, future studies could also include purely clinical factors to analyze the parameters that could be considered predisposing factors for access to LTC. For example, clinical parameters as blood pressure or glycemia could be helpful to investigate the profile of those who had access to LTC facilities. Finally, it would be necessary to enlarge the sample and make it more representative of the Italian population to extend it to other healthcare systems. On the other hand, one of the significant strengths of this paper is that this is, to our knowledge, the first paper that has reported the cumulative access rate to the LTC facilities.

Due to the complexity of this phenomenon, future studies could focus on the cause of LTC access. Still, more importantly, they could disseminate the LTC access rate for each country to investigate this phenomenon in more detail and at the same time compare the national factors involved. 

## 5. Conclusions

This study suggests that moderate frailty may be a significant predictor of LTC facilities among community-dwelling older adults as well as disability in some IADLs. Moreover, managing bio-psycho-social frailty could be essential in preventing adverse health outcomes, such as institutionalization, and a helpful tool in defining the care needs of community-dwelling older adults. Finally, the use of LTC facilities seems to be mainly determined by the imbalance between the perceived need for care and available support. Institutionalization has a meaningful impact on public health policy and costs: in order to meet the demand for care generated by older adults at a community level the implementation of Home Care Service is crucial.

## Figures and Tables

**Figure 1 healthcare-10-00317-f001:**
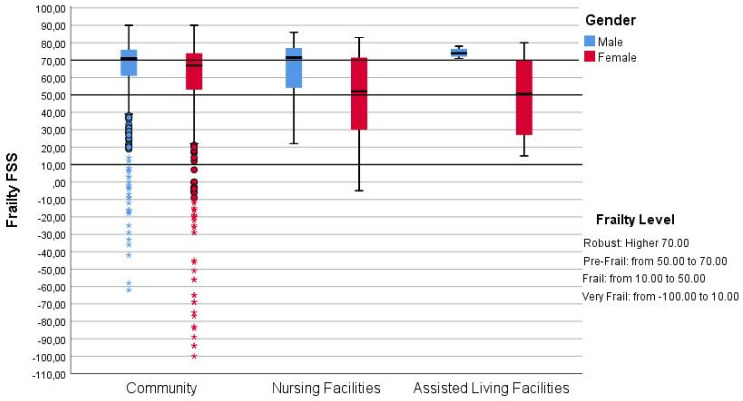
Box Plot of distribution of Frailty Final Synthetic Score at baseline, according to access to LTC facilities’ during the three-year follow-up and stratified by gender. Note. The score from −100.00 to 10.00 identifies the Very Frail older adults, a score between 10.00 to 50.00 the Frail older adults, a score between 50.00 and 70.00 the Pre-Frail older adults, and a score higher of 70.00 the Robust ones.

**Figure 2 healthcare-10-00317-f002:**
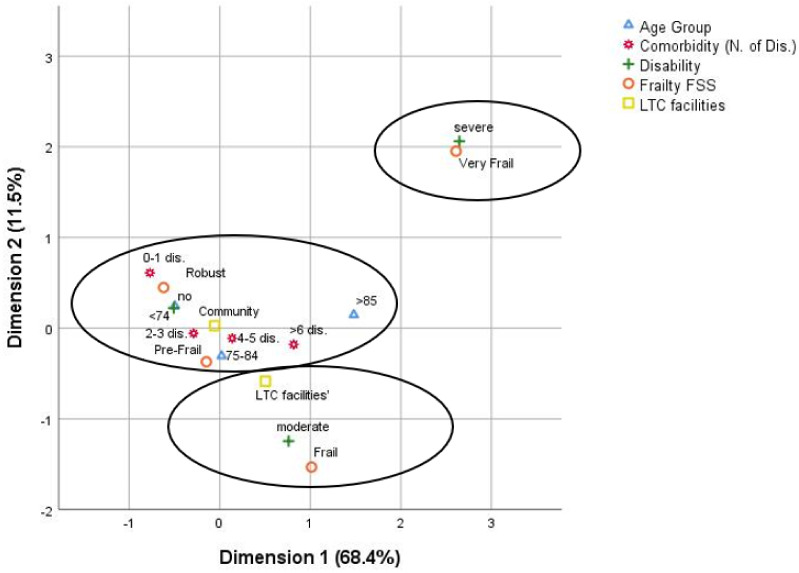
MCA dimension. Categorical variables: Age Group (<74 years, between 75 and 85, and >85); Comorbidity (0–1 dis., 2–3 dis., 4–5 dis, and >6 dis); Disability (no disability, moderate disability, and severe disability); Frailty FSS (Robust, Pre-Frail, Frail, and Very Frail); LTC facilities’ (community-dwelling older people and resident in Nursing Home Facilities or Assisted Living Facilities). Joint category plot of the explored variable categories. Dimension 1 explains 68.4% inertia and dimension2 explains 11.5% inertia. Coordinates are in standard normalization. Two-dimension of MCA solution was considered the most adequate: both presented an eigenvalue, 2.318 and 1.574, and inertia of 0.464 and 0.315 and Cronbach’s alpha, 0.711 and 0.457, respectively.

**Table 1 healthcare-10-00317-t001:** Main characteristic of the sample (N = 1224).

	N (%)	Mean ± SD	χ ^2^ *p*-Value
Gender Female	651 (53.2)		NS.
Age <74 years 75–85 years >85 years	540 (44.1) 511 (41.7) 173 (14.1)	76.25 ± 7.129	NS.
Cohabitants Alone Spouse Child Others Homeworker	253 (20.7) 638 (52.1) 267 (21.8) 40 (3.3) 26 (2.1)		NS.
Education No education Primary school Middle school High school Degree	94 (7.7) 577 (47.1) 305 (24.9) 187 (15.3) 60 (4.9)		NS.
Frailty Robust Pre-Frail Frail Very Frail	535 (43.7) 444 (36.3) 162 (13.2) 83 (6.8)	59.25 ± 27.96	0.001
Comorbidity Presence of two or more disease	1041 (81.7)		0.002
Disability No Moderate Severe	858 (70.1) 290 (23.7) 76 (6.2)		NS.

Note. The Pearson Chi-Square was calculated between the sociodemographic variables and the outcome variable, LTC facilities accesses. NS: Not statistically significant.

**Table 2 healthcare-10-00317-t002:** LTC facilities rate per 1000 observation/year stratifies for frailty level.

FRAILTY LEVEL	LTC Facilities Rate per 1000 Observation/Year	95%CI	
Robust	11.2	5.9	16.4
Pre-Frail	8.1	3.1	13.1
Frail	28.4	5.9	50.8
Very Frail	18.1	0.0	39.9

**Table 3 healthcare-10-00317-t003:** The absolute number of LTC access stratifies for frailty level and social support.

FRAILTY LEVEL		Without Social Support	With Social Support	Total
Robust	Community-Dwelling	62 (12.6)	429 (87.4)	491 (96.6)
	LTC facilities	1 (5.9)	16 (94.1)	17 (3.4)
Pre-Frail	Community-Dwelling	247 (61.8)	153 (38.3)	400 (97.5)
	LTC facilities	7 (70.0)	3 (30.0)	10 (2.5)
Frail	Community-Dwelling	89 (67.4)	43 (32.6)	132 (93.6)
	LTC facilities	6 (66.7)	3 (33.3)	9 (6.4)
Very Frail	Community-Dwelling	53 (74.6)	18 (25.4)	71 (95.9)
	LTC facilities	3 (100)		3 (4.1)

**Table 4 healthcare-10-00317-t004:** Multivariate Linear Regression Model.

						95% C.I per B	
	B	S.E.	β	t	*p*-Value	Lower	Higher
**FSS**	0.001	<0.001	0.273	4.800	<0.001	0.000	0.001
**Take medicine**	0.050	0.012	0.188	4.184	<0.001	0.027	0.074
**Inhabitants**	0.006	0.003	0.072	2.461	0.015	0.001	0.011
**Cardiovascular disease**	0.009	0.002	0.111	3.663	<0.001	0.004	0.014
**Managing money**	0.020	0.007	0.151	2.998	0.003	0.007	0.033
**Gender**	0.004	0.004	0.027	0.896	0.370 *	−0.005	0.013
**Age**	0.001	<0.001	0.051	1.577	0.115 *	0.000	0.001

Note. Dependent variable: LTC facilities’ rate per 1000 person/year. * Not statistically significant. Gender (Male coded as 0, Female coded as (1) and Age (continuous variable) as control variables. R2 = 4.8%, F = 8.15, *p*-value < 0.001. Continuous variable: FSS= Final Synthetic Score (higher value corresponds to low level of frailty, while low value corresponds to a higher level of frailty); Inhabitants = is the number of residents in the city where the data were collected (Higher value indicate the big city, while the low value is the small town). Dichotomous variable: cardiovascular diseases (0 coded as “Absence” and one coded as “presence”), take medicine, and managing money (0 coded as “Yes” and 1 coded as “No”).

## Data Availability

Not applicable.

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
