# Peer review of "Admission to the Long-Term Care Facilities and Institutionalization Rate in Community-Dwelling Frail Adults: An Observational Longitudinal Cohort Study"

_healthcare, 2022, doi:10.3390/healthcare10020317_

Round 1
Reviewer 1 Report
Thank you very much for the opportunity to review the manuscript.
This topic is interesting in that it provides necessary information regarding institutionalization rate of community-dwelling frail adults. However, as of current state of the manuscript, I unfortunately have to make the decision to reject the article. Some of the major reasons are listed below. I hope the comments will help the authors to improve the manuscript.
・Overall, there are major English language issues and extensive editing by professional editing service is required. Also, there are many mistakes in word use (eg. LINE 55: his → this?) that needs to be corrected.
・The logic of the story seems to be disrupted in many parts. For example, the last three lines in the abstract section is difficult to comprehend.
・The results require more accurate explanation. For example, Table 4 shows negative correlation of "Take medicine" and "Managing money". However, that trend can not be read from the results section. Furthermore, there are methodological issues such as not listing out the control variables. It would be interesting to see other variables that were insignificant and discuss the reason.
・MCA dimension result is interesting, but does not seem to be meaningfully used in the Discussion section.
・ The current Discussion section is difficult to understand and logical flow and story should be improved.
Author Response
This topic is interesting in that it provides necessary information regarding institutionalization rate of community-dwelling frail adults. However, as of current state of the manuscript, I unfortunately have to make the decision to reject the article. Some of the major reasons are listed below. I hope the comments will help the authors to improve the manuscript.
Thank you for taking the time to review this paper. We tried to address all your comments, as described in the dialogue below. We sincerely appreciate your insightful and constructive comments and suggestions. We believe that these have greatly strengthened the paper
・Overall, there are major English language issues and extensive editing by professional editing service is required. Also, there are many mistakes in word use (eg. LINE 55: his → this?) that needs to be corrected.
Our apologies for these grammar errors, the manuscript has been revised by an expert English-speaking colleague.
・The logic of the story seems to be disrupted in many parts. For example, the last three lines in the abstract section is difficult to comprehend.
Thank you, we appreciate this suggestion. We have modified the abstract to make it more readable.
Lines 23-26: Turquoise tracked change.
“ The results indicate that access to LTC facilities is not determined by severe disability, severe comorbidity, and higher frailty levels. Instead, it is related to moderate disability associated with a lack of social support. Therefore, the care policies need to enhance social interventions to integrate medical, nursing, and rehabilitative care.”
Moreover, we tried to make more understandable what have driven our thoughts in drafting the paper
・The results require more accurate explanation. For example, Table 4 shows negative correlation of "Take medicine" and "Managing money". However, that trend can not be read from the results section. Furthermore, there are methodological issues such as not listing out the control variables. It would be interesting to see other variables that were insignificant and discuss the reason.
Thank you, we appreciate this suggestion. We have repeated the analysis and reported the results, as more understandable to the readers. Lines 220-227 Turquoise tracked change.
“Table 4. Multivariate Linear Regression Model.
Note. Dependent variable: LTC facilities’ rate per 1000 person/ year. * Not statistically significant. Gender (Male coded as 0, Female coded as 1) and Age as control variable. R2=4.8%, F=8.15, P-value <.001. Continuous variable: FSS= Final Synthetic Score (higher value corresponds to low level of frailty, while low value corresponds to a higher level of frailty); Inhabitants = is the number of residents in the city where the data were collected (Higher value indicate the big city, while the low value is the small town). Dichotomous variable: cardiovascular diseases (0 coded as “Absence” and one coded as “presence”), take medicine, and managing money (0 coded as “Yes” and 1 coded as “No”).”
We also added in the results a couple of variables we consider relevant for the development of the paper (hospital admissions and ER accesses during the follow up): we observed that about half of the respondents who accessed the LTC facilities neither was admitted nor accessed the ER during the follow up. Lines 201-206 Turquoise tracked change
・MCA dimension result is interesting, but does not seem to be meaningfully used in the Discussion section.
Our apologies for not making this clear. We have added a paragraph in the discussion to explain it.
Lines 309-318 Turquoise tracked change.
“This is also evident from the MCA results, which showed the association between LTC facilities and moderate disability with a moderate level of frailty. Our results showed that, despite high levels of frailty and poor social support, the elderly population tends to stay at home as much as possible, looking for a balance between needs of care and available support, even if associated to a poor quality of life.
That is due to the complexity of this phenomenon, which may be influenced by social, economic, and cultural features related to the country where it is analyzed [40]. As a consequence of this complexity due to the multi-factorial influence, we have found that living in a big city increases the risk of institutionalization compared to a small town, perhaps thanks to the more availability of LTC structure or the poor livability in the city [46,47].”
・ The current Discussion section is difficult to understand and logical flow and story should be improved.
Thank you, we appreciate this suggestion. We have deeply modified the discussion to make it more readable.

Reviewer 2 Report
Abstract:
„Firstly, was to calculate the institutionalization rate“ This sentence should be revised.
Please provide more information on the sample in the abstract (e.g. sample size).
Are the institutionalization rate and the LTC facilities access rate the same? If yes, please use one of these terms consistently in the abstract.
Please provide information how long the observation period of the study was.
Introduction:
“only two reviews have analyzed the relationship between fragility and institutionalization“ Please stay consistent and use the term “frailty” instead of “fragility”.
Methods:
Please provide information on the possible range of the FSS.
“Then the Chi-Square was performed to evaluate the correlation 120 with the outcome variable“ Please specifically mention the outcome variable.
Please provide more information on how IADL and ADL disability level was determined.
Please already provide in the methods the information on the definition of the different age groups and comorbidity and on how cohabitants and education was assessed.
Results:
Table 1: Please provide the information in the legend for what the Chi-squared test was calculated.
Figure 1: Please define the bold lines in the legend.
Authors could think of providing a boxplot instead of a scatterplot, which might be more reader friendly.
Please provide the information on the definition of community-dwelling, nursing homes and assisted living facilities already in the methods.
Authors did not define social support. E.g. what is meant by lack of social support? Please clarify in the method section.
It remains unclear why the authors choose social support here to stratify the LTC access rate, why not education, age, comorbidities, etc.?
“if some variables significantly predict LTC facilties’ access rates. The selection of potential predictors remains unclear to the reader. Please provide more information on the selection process.
What other variables were included in this model, e.g. controlled for age, sex, etc.? What regression model was calculated?
Table 4 includes variables that were not mentioned before (e.g. managing money, take medicine). Where did this variables come from?
Author Response
Thank you for taking the time to review this paper. We tried to address all your comments, as described in the dialogue below. We sincerely appreciate your insightful and constructive comments and suggestions. We believe that these have greatly strengthened the paper
Abstract:
„Firstly, was to calculate the institutionalization rate“ This sentence should be revised.
Please provide more information on the sample in the abstract (e.g. sample size).
Are the institutionalization rate and the LTC facilities access rate the same? If yes, please use one of these terms consistently in the abstract.
Please provide information how long the observation period of the study was.
Thank you for your points. We have addressed all your comments about the abstract and we corrected the paper in order to be consistent through definitions and terms
Lines 13-26 Turquoise tracked change.
“The Worldwide aging and the increase of chronic disease impacted the Health System with an increased risk of admission to Long-Term Care (LTC) facilities for older adults. The study aimed to evaluate the admission rate to LTC facilities for community-dwelling older adults and investigate factors associated with these admissions. A secondary analysis of data stemming from an observational longitudinal cohort study (from 2014 to 2017) was performed. The sample made up by 1,246 older adults (664 females and 582 males, mean age 76.3, SD±7.1). The LTC facilities access rate was 12.5 per 1000 observations/ year. Multivariable Linear Regression identified frailty and cardiovascular disease as predictors of the LTC facilities' access rate. The Multiple Correspondence Analysis identified three clusters. The first includes those living at home with comorbidities, and the second is represented by frail people living in LTC facilities. Finally, the third was those with higher frailty but not linked with the LTC facilities. The results indicate that access to LTC facilities is not determined by severe disability, severe comorbidity, and higher frailty levels. Instead, it is related to moderate disability associated with a lack of social support. Therefore, the care policies need to enhance social interventions to integrate medical, nursing, and rehabilitative care.”
Introduction:
“only two reviews have analyzed the relationship between fragility and institutionalization“ Please stay consistent and use the term “frailty” instead of “fragility”.
Our apologies, now we use only frailty.
Methods:
Please provide information on the possible range of the FSS.
“Then the Chi-Square was performed to evaluate the correlation with the outcome variable“ Please specifically mention the outcome variable.
Please provide more information on how IADL and ADL disability level was determined.
Please already provide in the methods the information on the definition of the different age groups and comorbidity and on how cohabitants and education was assessed
All the requests have met, please see the tracked changes (green).
Results:
Table 1: Please provide the information in the legend for what the Chi-squared test was calculated.
Lines 155-156 Green tracked change.
“Note. The Pearson Chi-Square was calculated between the sociodemographic variables and the outcome variable, LTC facilities accesses. NS: Not statistically significant.”
Figure 1: Please define the bold lines in the legend.
We have added in the legend the definition of the bold lines.
Authors could think of providing a boxplot instead of a scatterplot, which might be more reader friendly.
We have modified the figure 1 as you suggest
Please provide the information on the definition of community-dwelling, nursing homes and assisted living facilities already in the methods.
Lines 89-92 Green tracked change.
“The Italian LTC facilities included the Nursing Homes (NH where social and health care is provided to partially or non-self sufficient patients) and the Assisted Living facilities (AL where only social services are provided to individual who should be self-sufficient/partially self sufficient at least at the time of the admission).”
Authors did not define social support. E.g. what is meant by lack of social support? Please clarify in the method section.
Lines 131-134 Green tracked change.
“Lack of Social Support has been defined as the presence of three conditions: living alone, lack of supporting neighbors, and lack of the possibility of receiving occasional help as delivering food or drugs at home in case of need. The last two conditions were self-declared by respondents (recorded by the FGE questionnaire). “
It remains unclear why the authors choose social support here to stratify the LTC access rate, why not education, age, comorbidities, etc.?
Lack of social support is one of the main reasons to access LTC. However, we choose to show in the table social support also because of the particular relationship with LTC access which was a biphasic one. Among the robust, LTC access was associated to satisfactory social support while it was exactly the contrary among the other ones. This means social support as a whole was not statistically associated to the LTC accesses, then it was excluded by any further analysis even if, in our opinion, its relationship with the LTC access was a matter of interest for the readers
“if some variables significantly predict LTC facilties’ access rates. The selection of potential predictors remains unclear to the reader. Please provide more information on the selection process.
Lines 138-143
“To deepen the analysis of factors associated with institutionalization, every single variable in the questionnaire, as well as each ADL according to Katz and IADL according to Lawton, have been matched with the LTC facilities access (dichotomous variable, Yes vs. No) and the Chi-Square was performed to evaluate the association. The variables that showed a statistically significant association with the LTC access (p<0.05) had been included in a multivariable linear regression model.”
What other variables were included in this model, e.g. controlled for age, sex, etc.? What regression model was calculated?
it was a Multivariable linear regression model. Only the variable showed in the analysis have been included in the model, because they were the ones related with statistic significance to the LTC access rate in the Univariate analysis
Table 4 includes variables that were not mentioned before (e.g. managing money, take medicine). Where did this variables come from?
These variables are part of the questionnaire for the assessment of ADLs and IADLs as reported in the methods.

Reviewer 3 Report
This is a large observational study that deals with a topic of increasing importance.
Since people become older frailty will become more and more relevant to single individuals and to the society as a whole. One main result of this study is the association between increasing severity of frailty and the growing risk of admission to a nursing home.
Such a relationship is relevant since frailty can be treated. Therefore, one consequence of this study might be to encourage practitioners and insurance-companies to screen community dwelling elderly individuals for frailty and to treat them accordingly in order to ameliorate frailty and to delaying nursing placement for some time.
Limitations of the study are discussed and results are compared to of other studies that deal with the issue of frailty.
Author Response
Thank you very much for your comments. We sincerely appreciate your commitment.
Round 2
Reviewer 2 Report
The authors appropriately addressed all my requests, and the revisions improved the manuscript substantially. I do not have any further comments.